# Cardiac Neural Crest and Cardiac Regeneration

**DOI:** 10.3390/cells12010111

**Published:** 2022-12-28

**Authors:** Shannon Erhardt, Jun Wang

**Affiliations:** 1Department of Pediatrics, McGovern Medical School, The University of Texas Health Science Center at Houston, Houston, TX 77030, USA; 2MD Anderson Cancer Center UTHealth Houston Graduate School of Biomedical Sciences, The University of Texas, Houston, TX 77030, USA

**Keywords:** neural crest cells (NCCs), regeneration, heart structure, cardiac regeneration

## Abstract

Neural crest cells (NCCs) are a vertebrate-specific, multipotent stem cell population that have the ability to migrate and differentiate into various cell populations throughout the embryo during embryogenesis. The heart is a muscular and complex organ whose primary function is to pump blood and nutrients throughout the body. Mammalian hearts, such as those of humans, lose their regenerative ability shortly after birth. However, a few vertebrate species, such as zebrafish, have the ability to self-repair/regenerate after cardiac damage. Recent research has discovered the potential functional ability and contribution of cardiac NCCs to cardiac regeneration through the use of various vertebrate species and pluripotent stem cell-derived NCCs. Here, we review the neural crest’s regenerative capacity in various tissues and organs, and in particular, we summarize the characteristics of cardiac NCCs between species and their roles in cardiac regeneration. We further discuss emerging and future work to determine the potential contributions of NCCs for disease treatment.

## 1. Introduction

The heart is a muscular and complex organ whose vital role includes the circulation of blood and nutrients throughout the biological system. In contrast to other species, the mammalian heart is composed of four chambers (right atria, left atria, right ventricle, and left ventricle), with various connecting vessels and arteries, including the aorta and pulmonary vessel. The mammalian heart is derived from numerous cell types, including the first heart field, second heart field, and neural crest (NC) population. The NC is an embryonic stem cell population known for its temporary migratory potential and multipotency, specific to vertebrate development. Neural crest cells (NCCs) are derived from the neural plate border during neurulation, while simultaneously undergoing epithelial-to-mesenchymal transition, a process that shifts cells into a mesenchymal state encompassed with enhanced migratory potential [1,2]. Based on the initial axial position and site of contribution, NCCs are divided into specific subpopulations: cranial, vagal, trunk, and sacral [3]. Although the vagal NC mainly contributes to the development of the enteric nervous system, a small number of cells, deemed the cardiac NC, contribute to the development of the cardiac system [4,5]. It is currently perceived that the first and second heart field contribute to the ventricles and atria, and the NC mainly contributes to the cardiac valves, interventricular septum, and both the aorta and pulmonary vessel [3,6,7]. However, recent investigations have focused on determining the contribution of cardiac NCCs to various portions and locations of the heart. Comparably, cardiac NC contribution, although varying between model systems, has shown promising results for contributing to the myocardium and assisting in the regenerative capacity of cardiac tissue in zebrafish (*Danio rerio*) [8,9]. This potential regenerative capacity to cardiac tissue in amniotes poses interesting avenues to advance the treatment of various cardiac diseases. Heart disease is currently the leading cause of death in the United States. Ranging in type and severity, the damage caused to the heart results in either death or irreplaceable damage to the function and/or structural integrity of the heart. Therefore, determining the contribution and regenerative capacity of the cardiac NC population in mammalian systems is of high clinical significance. 

## 2. Cardiac Neural Crest Contribution to the Heart between Species

The ablation or disruption of cardiac NCCs during embryogenesis can result in congenital heart defects (CHDs), the most common human birth defect, presenting as various cardiac abnormalities that can affect both the structure and the function of the heart [3]. Although phenotypes between biological models are comparable, to an extent, the contribution of the cardiac NC to the heart varies between species and may be a contributing factor that impacts the functional possibilities of the cells for regeneration later in the adult system. 

### 2.1. Chick/Quail-Chick Chimera

The contribution of the NC to cardiac development has been heavily studied using chick or quail-chick chimera models. In the quail-chick chimera, ablation of the NC from somites 1 through 3 prior to migration resulted in aorticopulmonary defects, along with alterations of the outflow tract (OFT), leading to rarer CHD phenotypes, such as transposition of the great arteries [4]. Furthermore, ablation of NCCs destined for aortic arches 3, 4, and 6 in the chick have been shown to result in persistent truncus arteriosus (PTA), aortic arch defects, and incomplete looping of the heart [10,11]. Similarly, ablation of pre-migratory NCCs destined for aortic arches 3 and 4 presented with cardiac alterations, including changed stroke volumes, reduced cardiac output, and decreased ejection fractions [11,12]. These results show not only that cardiac NCCs are vital for proper heart development in the chick, but also that this cell population may be needed for the cardiac conduction system, a vital component necessary for electrical contraction in the heart. More recently, Tang and colleagues identified that with the use of replication-competent avian retroviruses to mark NC progenitors for lineage analysis in chicks, cardiac NCCs contributed to the trabecular myocardium of the ventricles, a previously unreported contribution of NCCs [13]. Previously, it was accepted that cardiac NCCs only contributed to portions of the aorta and pulmonary vessels and valves, along with the membranous interventricular septum [4,14,15]. However, this identification of NCCs in the ventricles provides a novel insight into NC contribution and leads to further questions regarding their function and contribution throughout development and regarding regeneration capability. 

### 2.2. Mouse

Not long after the identification of cardiac NCCs in chicks, the mouse (*Mus musculus*) was identified to have similar contributions of NCCs to the heart. As a fellow amniote, the cardiac structure of the mouse is four-chambered, consisting of two atria and two ventricles, similar to that of the chick. It was corroborated that cardiac NCCs in the mouse also delaminate to pharyngeal arch arteries 3, 4, and 6, which will then further migrate to the heart [16]. Similar to recent evidence found in chicks, a growing number of studies also suggested the ability of mouse NCCs to differentiate into cardiomyocytes [13,17,18,19,20,21], yet conclusions may vary using different NCC lineage-tracing mouse models [22,23,24]. 

Tang et al. examined mice that had NCCs marked using cytoplasmic GFP (*Wnt1-Cre;ZsGreen^fl/fl^* and *Wnt1-Cre2+;R26mTmG)*) and found that at embryonic day (E) 15.5, a large number of cells in the OFT, interventricular septum, and myocardium of the ventricles were NC positive [13]. The authors do note that the number of *Wnt1-Cre*-positive cells remains stable postnatally and does not undergo cell division or apoptosis, providing similar information that has been noted in both chicks and zebrafish, indicating an evolutionary role of the NC within these cardiac regions [13,25,26]. The finding of the NC population in both of the mouse ventricles warrants further investigation into the NC-specific function in these regions. Furthermore, the stability of the NC population raises interest regarding the potential ability to contribute to cardiac remodeling due to injury, including regeneration. 

### 2.3. Zebrafish

In contrast to amniote models, the zebrafish (*Danio rerio*) has been a driving model in studying development, as its translucent body allows for clear visualization of fluorescence-labeled NC contribution [27,28]. Unlike the structure of the amniote heart, zebrafish hearts are tubular in structure, consisting of one atrium and one ventricle, and maintain a one-directional flow [26,29,30]. As the developing heart in zebrafish varies compared to amniote models, this suggests that NC contribution varies as well. In contrast to the mouse or chick, the zebrafish heart does not consist of a ventricular septum due to having a single ventricle, clearly indicating varying cardiac NC contribution [26,30]. Sato and Yost found that in zebrafish, cardiac NCCs contribute to the bulbus arteriosus, ventricle, atrioventricular junction, atrium, and muscle formation in the myocardium [25]. The Kirby group also found that by using cell marking, cardiac NCCs migrated to the myocardial wall of the heart tube, and laser ablation of the pre-migratory cardiac NCCs resulted in the loss of NC migration to the heart and failed heart looping, along with reducing the ejection fraction and cardiac output [26]. Although it has further been corroborated that NCCs integrate into the myocardium in the zebrafish heart, the Chen group found that NCCs in the zebrafish can be characterized into two populations: one that gives rise to cardiomyocytes and another that contributes to the endothelium and bulbus arteriosus [8]. Furthermore, the Chen group found that ablation leads to reduced heart rates, defective myocardial maturation, and failure to recruit progenitor cells from the second heart field, indicating the need for further investigation into the cell–cell communication between the various cardiac contributing cell populations [8]. 

### 2.4. Frog

Different from chicks, mice, and zebrafish, the frog (*Xenopus laevis*) heart is comprised of three chambers (two atria and one ventricle) containing incomplete OFT septation, which is referred to as a spiral septum, that assists in directional blood flow [31,32,33]. However, although structurally different, studies have also investigated the variation and/or conservation of the NC population during heart formation. Studies have confirmed that in the frog, the NC population contributes to the development of the aortic sac and arch arteries, and it does not contribute to the OFT cushion mesenchyme, which is solely contributed by the second heart field, similar to the cushion composition in mice and chicks [32,34,35]. This observation that NCCs contribute to the cardiac cushions in humans, mice, and chicks can be considered a characteristic of a higher vertebrate species, as NCCs do not populate the cardiac cushion in frogs, purposing that the recruitment of cardiac NCCs into the OFT cushions allows for an increase in cells in the OFT septum to complete septation [34]. Furthermore, using orthotopic translation in frog chimeras to lineage trace NCCs, it was found that a smaller portion of NCCs was discovered in the wall of the truncus arteriosus, providing evidence that although various abilities of the NC are similar among species, further investigation is needed into the effect of such variations [35]. 

NC ablation studies in frogs have produced varying results. For example, Martinsen et al. ablated NCCs from the cranial to mid-trunk level during embryogenesis, which resulted in abnormal cardiac development, including an elongated and un-looped heart tube, pericardial edema, and lack of normal heart tube formation, meaning that NCCs are required for normal cardiac development in the frog [36]. However, other studies have found that the ablation of pre-migratory NCCs did result in the loss of the aortic sac and arch arteries, but presented with normal spiral septum formation, suggesting that in the frog, the septum is derived from the second heart field and not the NC [34]. This variation between cardiac alterations suggests high crosstalk with the second heart field, as seen in mice and chicks, but warrants a more concise conclusion into the potential abilities of the NC to crosstalk with other cell populations [37,38,39,40]. 

## 3. Regenerative Capacity of the Neural Crest

Although NC contribution is most commonly investigated during embryogenesis, recent studies have begun to determine this population’s potential for regeneration at later stages. It is most commonly accepted that NCCs are multipotent early in development, but have the possibility to lose their multipotency once differentiated into various cell types, based on the area of contribution. Tissue regeneration is a vital and emerging ability by which the structure and function of damaged tissues and organs can be restored. The potential contribution of the NCC lineage to regeneration could offer great potential to regenerative medicine and disease modeling; however, this ability is not yet fully understood in various tissues and between different species. 

### 3.1. Gastrointestinal Tract and Enteric Nervous System 

One field of interest that has sparked an investigation into NC regenerative capacity is the gastrointestinal tract. In particular, the intestine has one of the highest regenerative capacities in the human body [41]. The coordination of the gastrointestinal tract is regulated by the enteric nervous system (ENS), a network comprised of neurons and glial cells that arise from the NC [42,43]. To understand the contribution of the NC to this highly regenerative area, Kruger and colleagues investigated the gastrointestinal tract in adult rats and found the persistent existence of NCCs. These postnatal NCCs were able to self-renew extensively in culture, but were overall not as extensive as NCC regeneration of the fetal gut [44]. Furthermore, the authors were able to determine that the NCCs of the adult gut were still active and able to give rise to neuro-transmitting neurons but were unable to create certain neural subtypes that were capable of being produced by the fetal gut [44]. They suggest that this reduction in producing various neuronal subtypes involves a loss of BMP expression but an increased response to gliogenic factors at postnatal stages [44]. Although the complete functional significance of NCCs in the adult mammalian system is still unknown, these findings of NCCs in the adult gut suggest new possibilities for NCCs in regeneration. 

Recently, Yuan and colleagues have investigated the regenerative capacity of the ENS and whether NCCs, other than those of the enteric NC, have the ability to regenerate functional enteric neurons and neurons of the intestine [45]. Previous studies have shown that NCCs from the native intestine and ENS are capable of differentiating into functional enteric neurons and were able to rescue bowel motility [46,47,48,49,50]. For example, Cooper et al. engrafted enteric NCCs into the mouse gut and found that enteric neurons and glia arose and were maintained over two years without presenting long-term complications, providing invaluable contributions to regenerative therapies [48]. However, Yuan et al. investigated whether, in the adult mouse, trunk pre-migratory NCCs that do not contribute to the ENS are also capable of generating enteric neurons [45]. They determined that trunk NCCs transplanted into the colon of adult mice were able to form neuronal networks, including enteric neurons and ganglion-like anatomy in the colon, suggesting that non-intestinal pre-migratory NCCs were able to establish neuron-like characteristics and can successfully integrate into the intestine [45]. 

### 3.2. Cranial Bones, Bone Marrow, and Teeth

One major contribution of the NC is to assist in cranial skeletal formation during embryogenesis, including cartilages and bones of the head [51,52,53,54]. Beyond cranial bones, studies have also indicated that NCCs contribute to bone marrow and tooth formation [55,56]. However, whether the NC can re-activate stem cell-like properties or NCCs maintain multipotency postnatally in such structures is still under investigation. 

Although NCCs give rise to the majority of bone, cartilage, and connective tissues of the skull, little is known about the regenerative ability of the NCC lineage in cranial structures after injury or disease. However, regarding the reactivation of stem cell-like characteristics in adult bones, Ransom et al., with the use of a mandibular distraction osteogenesis mouse model, identified that after injury, NC development-related genes, including *sox10, sox18, and elk3,* were upregulated within newly forming bones of the jaw [57]. Furthermore, it was identified that post-migratory cranial NCCs were not only self-renewing and able to form bone matrix in culture, but that subcutaneous transplanted post-migratory cranial NCCs in mice were able to regain their ability to differentiate into osteocytes and adipocytes, along with assisting in calvaria bone formation [58]. However, the ability and corresponding mechanisms of cranial NCCs to re-active or maintain their multipotency within bones of the skull and to properly contribute to new bone formation have yet to be deciphered. 

Bones, including portions of the skull, are composed of compact and cancellous tissue, along with bone marrow at the core. Bone marrow is comprised of red blood cells, white blood cells, and platelets, along with a large portion of stem cells. The origin and differential capability of such stem cells is an area currently under investigation. For example, Isern and colleagues identified that mesenchymal stem cells from the marrow of long bones arise from trunk NCCs, and their contribution to marrow and surrounding bones may be correlated to nestin expression [59]. Furthermore, Nagoshi et al. also identified NC-like stem cells in the bone marrow, and although regeneration of marrow was not confirmed, they determined that cells collected from the marrow of mice tibias and femurs at various postnatal stages were able to differentiate into neurons, myofibroblasts, and glial cells, posing novel advances toward the ability of NC-derived bone marrow to contribute to bone regeneration [56].

Similar to bones, teeth, which are located in the mandible, are a known structure contributed by NCCs during development [60]. A number of groups have begun to decipher the regenerative capacity of NCCs that contribute to tooth formation. For example, Zhang et al. recently showed that NCCs in vitro (O9-1 mouse NCC cell line) are able to differentiate into odontoblasts, and that in vivo, both primary O9-1 cell scaffolds and induced pluripotent stem cells (iPSCs) were able to contribute to dentin-pulp regeneration of the mouse tooth [61]. Furthermore, Chung et al. also identified that transplanted tooth germs with post-migratory cranial NCCs contributed to tooth formation and survival of the tooth germ, possibly due to BMP signaling through the regulation of Smad4 [58]. Although not comprised of bone or bone-like components, mammalian palatal tissue is highly regenerative and contributed significantly by NCCs [62,63,64,65]. Studies have so far identified regions of the palate that consist of NC-like stem cells that maintain proliferative and differentiation abilities, possibly providing avenues into regenerative disease therapy for defects of both the hard and soft palate [66,67]. 

### 3.3. Peripheral and Central Nervous System

The formation of glial and neuronal cell lineages is a process that is assisted by the NC population during development. During development, NCCs from the trunk region give rise to numerous sub-lineages, including glial cells of the peripheral nervous system (PNS). Glial cells contribute to the structure of both the PNC and central nervous system (CNS), assisting in the protection and regulation of neurons, particularly through NCC-derived Schwann cells [68]. The regeneration of nerve fibers and their supporting cells has been a standing field of interest regarding functional recovery to structures such as the spinal cord after injury. However, to date, success in nerve and glial regeneration has been minimal, but recent work has shown the potential of NCCs in regeneration. 

Recent advances regarding PNS regeneration have been made possible due to the use of NC-like stem cells derived from human embryonic stem cells or iPSCs. For example, Huang and colleagues used NC-derived iPSCs to construct nerve conduits that, when implanted into a rat sciatic nerve transection model, were able to increase functional nerve recovery [69]. Supporting studies using sciatic nerve defect models and NC-derived human iPSCs found that in vivo, grafted cells proliferated and successfully migrated throughout the conduit after transplantation [70]. Furthermore, Kimura et al. discovered that grafted NC-derived iPSCs also contributed to the increased strength of the leg muscle, indicating functional recovery of the sciatic nerve after injury [70]. Similar conclusions have been made by other groups, indicating that NC-derived iPSCs or NC-derived embryonic stem cells are a valuable tool that can contribute to nerve regeneration. However, the mechanisms of this ability have yet to be determined [71,72,73]. 

Although sciatic nerve models have provided valuable information regarding the contribution of NCCs to nerve regeneration, others have begun to apply NC-derived stem cells to spinal cord injuries [74,75]. Similar to the PNS, the CNS is also composed of various types of glial cells and neurons; however, the CNS is mainly composed of the spinal cord and the brain, both of which are contributed highly by the NC population. In the effort to determine whether NCCs are capable of contributing to CNS regeneration and repair, Saadai et al., using human iPSCs from skin fibroblasts, seeded differentiated NCCs into fetal lambs with spina bifida (an open neural tube defect) and found that iPSCs NCCs produced neuronal lineages shown through the expression of the mature axonal neurofilament marker NFM [74]. More recently, Jones et al. investigated whether human embryonic stem cell-derived NCCs could be used as a therapeutic for adult spinal cord injury and discovered that such derived NCCs were able to differentiate and stimulate neuronal growth both in vitro and in vivo, accompanied by forelimb function recovery in their rat spinal cord injury model, warranting further investigation into the mechanisms and functional capabilities of NC-like stem cells for therapeutic treatments [76]. 

Regarding the brain, questions have been raised as to whether NC-derived cells contribute to either functional or structural regeneration after injury. A common injury to the brain is stroke, in which damage occurs due to an interruption of blood flow. To date, multiple groups have reported that NC stem cells contribute to pericytes of the CNS, and that ischemia-induced stem cells are contributed by NC-derived pericytes after stroke [77,78]. Furthermore, studies have revealed that the transplantation of NCCs from bone marrow or the epidermis to the corpus callosum in a lipopolysaccharide-induced inflammatory lesion rat model indicated that these NCCs were able to migrate toward the lesion, where they remained for several months, during which time glial cell fates were adopted, raising the question as to how signaling from an injured area attracts and communicates with surrounding NC-like stem cells to assist in recovery [79].

However, the ability of NC-derived cells in vivo to maintain or re-active their multipotency is a field that has yet to be defined regarding both the CNS and PNS, but the conclusions made by NC-derived stem cells pose interesting avenues to the potential re-activation of a NC-like state. 

## 4. Cardiac Neural Crest in Cardiac Regeneration

During early mammalian development, the heart maintains its regenerative capacity; however, shortly after birth, this ability is lost. Postnatal cardiac progenitors remain a challenging and controversial issue in the cardiac field. Recent studies have begun to investigate the potential ability of the heart to re-activate regenerative ability, particularly through the NC, to assist in cardiac regeneration after injury. 

It has previously been established that zebrafish maintain regenerative abilities throughout their systems, including the fins and retina [80,81,82,83]. Furthermore, it has been identified that adult zebrafish hearts are able to regenerate ventricular myocardium, without scarring, through cardiomyocyte dedifferentiation and proliferation [84,85]. However, until recently, it was unknown whether the NC population assists in this regeneration capacity of the heart. In addition to the established NCC contributions to cardiovascular development, numerous groups recently determined that NCCs in the zebrafish heart also contribute to the cardiomyocyte population [9,13,86]. Based on this, Tang and colleagues further investigated whether the NC population of the zebrafish heart also plays a role in cardiac regeneration. Using a *sox10* promoter, expressing GFP (*Tg(-4.9sox10:eGFP*) to label embryonic NCCs, it was found that though *sox10* expression is down-regulated after NCCs reach the heart, the removal of a portion of the adult ventricular apex stimulates *sox10*-GFP expression, along with the NC marker *tfap2a*, in cardiomyocytes near the injury site, suggesting the reactivation of a NC-like state for cardiac regeneration [13] (Figure 1). Furthermore, Sande-Melón et al. determined that pre-existing *sox10*-positive NCCs not only contributed to the zebrafish adult heart, but that after ventricular cryoinjury, the number of *sox10*-expressing cells significantly increased in the myocardium near the injured area [9] (Figure 1). 

Although regeneration in zebrafish has shown promising roles for NCCs in cardiac regeneration, less is known about the contribution of the NC during mammalian cardiac regeneration. Similar to zebrafish, it was identified that NCCs are present in the postnatal mouse heart and can differentiate into cardiomyocytes [17,18,19,20,21]. Tamura and colleagues found that after myocardial infarction in adult mice, GFP-expressed NCCs migrated to the border of the infarcted region and were able to differentiate into cardiomyocytes, contributing to the regeneration of the myocardium [19]. The authors suggest that this migration of NCCs after myocardial infarction is due to monocyte chemoattractant protein-1 (MCP-1) expression in the infarcted area that provides guidance cues to NCCs [19] (Figure 1). In contrast, although Hatzistergos and colleagues found that a population of NCCs generate cardiomyocytes postnatally, these cells were not proliferative and had lost their regenerative capacity [17].

## 5. Conclusions and Future Perspectives 

NCCs are a multipotent cell population that are active during vertebrate embryogenesis and can contribute to numerous portions of the developing system. It is well accepted that cardiac NCCs contribute to the OFT, great vessels, and septa of the heart. Furthermore, in mammals, it is understood that the heart loses its regenerative capacity shortly after birth. However, recent studies have begun to pose various insights into the regenerative capability of the heart and the contribution of the NC to such ability. 

The use of various model organisms has provided vital information on how NCCs contribute to cardiac formation, and more recently, these have been providing insight into the possible contribution of this cell population beyond that which was previously accepted. Although progress has been made in understanding the function of the NC between species, cardiac formation, and correspondingly, NC contribution, varies between model systems, which poses questions as to the evolutionarily conserved abilities of the NC. For example, the finding that NCCs differentiate into cardiomyocytes and assist in the reformation of the ventricular myocardium has been clearly shown in adult zebrafish [9,13,86]. However, the confirmation of this ability in mammalian species has proved more challenging, but recent advances have shown potential corroboration of such ability [17,19,20,21,87]. This variation in the capacity of NCCs in different species may be due to the variation in how NCCs contribute to cardiac formation. The location of NCCs in the heart during embryogenesis could pose an advantage in certain species, as the recruitment of NCCs to the injured area may lead them to be able to receive certain cues from surrounding cells to re-instate stem cell-like properties. Furthermore, the axolotl should be additionally investigated into the NC’s role in regeneration. The axolotl is a type of salamander with an astounding capacity for regeneration of its system, including the tail and limbs, along with portions of its spinal cord and brain [88,89,90,91,92,93]. As NCCs contribute to a large portion of various developmental processes, it only raises the question as to whether NCCs contribute to the regenerative ability of the axolotl. Multiple groups have reported the ability of the axolotl to regenerate cardiac tissue without scarring after partial ventricular amputation assisted by cardiomyocyte proliferation [94,95], with Cano-Martinez et al. further indicating that the axolotl is also able to restore cardiac function after injury [94]. To date, little is known about the NC’s contribution to the development of the axolotl heart [35,96]. However, the current data pose an interesting avenue into the potential abilities of the NC regarding such regenerative capacity, which could lead to advances in cardiac NC regeneration in mammalian species and advances in cardiac disease therapies. 

As the contribution of NCCs to the heart varies between model systems, it should be further investigated whether regulatory networks controlling NC response to injury and tissue regeneration also differ. For example, in zebrafish, it was identified that *sox10* is potentially a vital component in cardiac NC reactivation and cardiomyocyte proliferation [13,86]. However, this role for *sox10* has yet to be investigated and corroborated in other vertebrate systems, such as the mouse or chick. Although there are various genes and networks that are staples in cardiac mechanism studies, such as BMP and Wnt [44,97,98,99], one signaling pathway that warrants further investigation regarding the NC’s contribution to cardiac regeneration is the Hippo signaling pathway. The Hippo signaling pathway is a highly conserved regulator of organ size and tissue growth. Previous studies have reported that during mouse embryogenesis, the deletion of Hippo pathway core kinases promotes cardiomyocyte proliferation, and that constitutively active Yap, a downstream effector of the Hippo pathway, increases cardiomyocyte proliferation and heart size both in the embryonic and postnatal mouse [100,101]. To date, numerous studies have identified Yap as a vital factor in regulating cardiomyocytes and neonatal cardiac regeneration, and deficiencies in Yap or its downstream targets, such as *Wntless*, result in increased scar size and fibrosis after myocardial infarction [98,102]. Furthermore, it was identified that overexpression of Yap (YapS112A) after injury at postnatal day 28 reduced fibrosis, increased myocardial tissue, enhanced cardiac function, and promoted cardiomyocyte proliferation and survival, indicating that Yap is vital for postnatal-cardiac regeneration after injury in mice [102]. As NCCs have been shown to give rise to cardiomyocytes among various species, further investigation into the Hippo signaling pathway’s role in regulating NC-derived cardiac tissue regeneration could pose a novel mechanism and role for both Hippo signaling and cardiac regeneration. 

To enhance current therapies for patients with heart injuries, numerous studies have begun investigating the ability of NC-derived stem cell models. As discussed, both human embryonic stem cells and iPSCs are currently at the forefront of regeneration strategies [61,69,70]. Although results are promising, further investigations are needed into the potential of NC-like stem cells in specific tissues to contribute to regeneration. Promising results have been shown in the ENS [44,45], but similar abilities of NCCs in the cardiac system have not yet been identified. Further investigations should include whether NCCs from embryonic stages remain multipotent in a dormant stage, or whether such cells maintain specialized fates and later dedifferentiate to contribute to regeneration upon injury.

This review summarizes the contribution of the cardiac NC to the heart between species and the possible contribution to regenerative capacity. Furthermore, we have discussed recent advances in the field of cardiac regeneration, with emphasis on the investigation into how NCCs may be a pivotal cell population that could provide novel information regarding tissue regeneration therapies. Despite the recent advances in understanding NC-derived cardiac regeneration, many questions still persist regarding the regenerative capacity of the adult mammalian heart and the respective mechanisms. Research is also needed to determine the evolutionarily conserved elements of NCs in the cardiac system in order to better understand the abilities and functional contributions of the NC.

## Figures and Tables

**Figure 1 cells-12-00111-f001:**
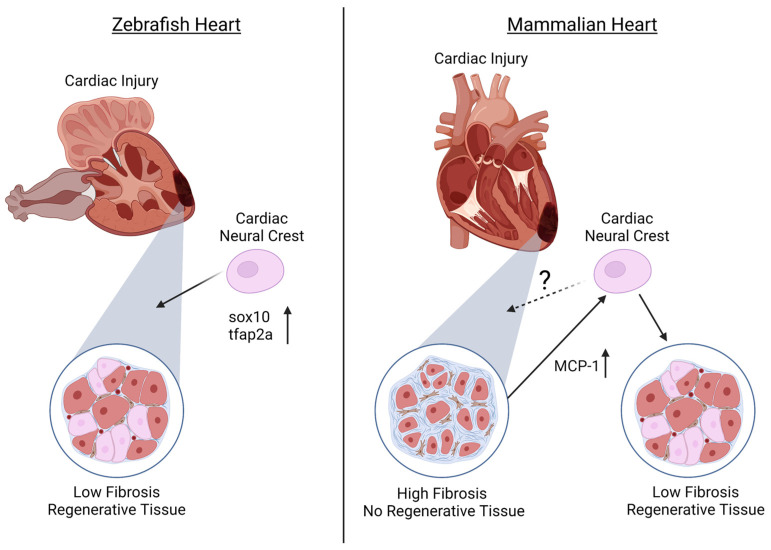
The ability of the cardiac neural crest (NC) to contribute to regeneration after cardiac injury in zebrafish and mammalian hearts. In the zebrafish, injured ventricular tissue has been shown to populate a large number of NC-positive cells that express high levels of *sox10* and *tfap2a* that give rise to cardiomyocytes and myocardium of the regenerated ventricle. In the mammalian heart, the contribution of the NC to the ventricle and their ability to regenerate or maintain a stem cell-like state is still unknown. Although mammalian ventricular cardiac injury results in scarring and fibrosis with little-to-no regenerative ability, there is potential that the injured tissue releases the chemokine MCP-1, signaling to cardiac neural crest cells (NCCs) to migrate to the injury site to assist in tissue regeneration of the ventricle. (Created with Biorender, accessed on 29 November 2022).

## Data Availability

Not applicable.

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
