# Peer review of "Cardiac Neural Crest and Cardiac Regeneration"

_cells, 2022, doi:10.3390/cells12010111_

Round 1

Reviewer 1 Report

This is a review article by Erhardt and Wang discussing the potency of cardiac neural crest cells from different vertebrate species and the regenerative capacity of neural crest cells to cells in different tissues including the cardiomyocyte, which is critical for the repair post cardiac infarct. In general, I find the review well-organized and comprehensive.  I do noticed that some studies relevant to the multipotency or regenerative capacity of neural crest cells were missed in section 3. I would recommend the authors consider adding some of them to make the summary more complete.

3.1. GI system: Joesph NM et al., 2011 described NC derived glial cells in adult rodent gut. 

3.2. Bone: NC has been reported to contribute to cells in the bone marrow, which may contribute to the differentiation of different cell lineages postnatally. Isern J et al., 2004; Nogoshi N et al., 2008.

Also, NC derived stem cells have been identified in palatal tissues. Widera D. et al., 2009; Zeuner MT et al., 2018.

3.3. PNS: The authors emphasized on the use of NC-derived iPSCs here, but it should be noted that NC-derived cells, such as Schwann cells have the potential in nerve repair after injury, which has been reviewed by Jessen KR et al in 2019.

Additionally, pericytes have been reported to give rise to neuronal and vascular cells in the CNS. Nakagomi T et al., 2015; Tatebayashi K et al., 2017.

Author Response

We thank the reviewer for the comments and suggested studies. For the GI system, we have gone ahead and added the listed study to this section. Regarding the Bone section, we have revised this section to include cranial bones, bone marrow, and teeth to provide a well-rounded review. We have included the suggested studies in this section accordingly. Lastly, for the Peripheral Nervous System section, we have also broadened the title to Peripheral and Central Nervous system as our previous version did include aspects of the central nervous system but this was not made clear. We have also included the addition of the suggested studies in this section. 

Reviewer 2 Report

A nice and timely review on an interesting topic regarding the contribution of neural crest cells in cardiac regeneration. It's nicely written.

Author Response

We thank the reviewer for their comments regarding the preparation of this review.